# Impact of co-morbid common mental disorder symptoms in people with epilepsy in Ethiopia on quality of life and functional disability: a cohort study

## Research Article

epilepsy; depression; disability; low income country; Africa

**Corresponding author:**
Ruth Tsigebrhan;
Email: r_tessera@yahoo.com

Ruth Tsigebrhan[1,2] (iD), Girmay Medhin[1,3], Merga Belina[1], Charles R. Newton[4,5] and Charlotte Hanlon[1,6] (iD)

[1]Centre for Innovative Drug Development and Therapeutic Trials for Africa (CDT-Africa), College of Health Sciences, Addis Ababa University, Addis Ababa, Ethiopia; [2]Department of Psychiatry, WHO Collaborating Centre in Mental Health Research and Capacity-Building, School of Medicine, College of Health Sciences, Addis Ababa University, Addis Ababa, Ethiopia; [3]Aklilu Lemma Institute of Pathobiology, Addis Ababa University, Addis Ababa, Ethiopia; [4]Neuroscience Unit, KEMRI-Wellcome Trust Research Programme, Kilifi, Kenya; [5]Department of Psychiatry, University of Oxford, Warneford Hospital, Oxford, UK and [6]Division of Psychiatry, Centre for Clinical Brain Sciences, University of Edinburgh, Edinburgh, Scotland

## Abstract

The objective of this study was to investigate the impact of common mental disorder (CMD; depression/anxiety) symptoms and risky substance use in people with epilepsy in Ethiopia (four districts) on quality of life (QoL) and functioning over 6 months. A prospective cohort study was carried out. Multivariable linear regression followed by structural equation modelling (SEM) was employed. In the multivariable regression model, neither CMD symptoms (β coef. = −0.37, 95% confidence interval [CI] −1.30, +0.55) nor moderate to high risk of alcohol use (β coef. = −0.70, 95% CI −9.20, +7.81) were significantly associated with a change in QoL. In SEM, the summative effect of CMD on QoL was significant (B = −0.27, 95% CI −0.48, −0.056). Change in functional disability was not significantly associated with common mental disorder (CMD) symptoms (β coef. = −0.03, 95% CI −0.48, +0.54) or with moderate to high risk of alcohol use (β coef. = −1.31, 95% CI −5.89, 3.26). In the SEM model, functional disability was predicted by both CMD symptoms (B = 0.24, 95% CI 0.06, 0.41) and seizure frequency (B = 0.67, 95% CI 0.46, 0.87). In this rural Ethiopian setting, co-morbid CMD symptoms and seizure frequency independently predicted functional disability in people with epilepsy.

## Impact statement

This study highlights the significant impact of co-morbid common mental disorders (CMDs) and seizure frequency on functional disability in people with epilepsy (PWE) in a rural Ethiopian setting. It also emphasises the complex interplay between CMD symptoms, substance use and functional disability in PWE. Our findings demonstrate that while direct associations between these factors may not always be apparent in traditional regression analyses, structural equation modelling can identify significant indirect effects. These results underscore the importance of comprehensive care that addresses both mental health and epilepsy-related factors to improve the overall well-being of individuals living with epilepsy. By identifying these complex relationships, this research provides valuable insights for healthcare providers and policymakers in developing targeted interventions and improving the quality of life for PWE.



## Introduction

Substantial global evidence indicates that there is a high level of co-morbid common mental disorders (CMDs), especially depression and anxiety, among people with epilepsy (PWE) compared to the general population (Doherty et al., 2022; Muhigwa et al., 2020; Mula et al., 2022). CMDs include the presence of combination of nonspecific depressive, anxiety and somatic symptoms (Hanlon et al. 2008c). The pooled prevalence of anxiety disorders in a meta-analysis of 69 studies in adults with epilepsy was 21.7% (95% confidence interval [CI] 19.2–24.3%), and the prevalence of co-morbid depression (meta-analysis of 95 studies) was 18.9% (95% CI 15.5–22.3%; Doherty et al., 2022). The pooled prevalence of comorbid alcohol abuse from a meta-analysis of seven studies was 5.6% (0.5–8.7%) and drug abuse was 6.1% (0.6–20.6%; Lu et al., 2021). The odds of these CMDs were significantly increased in PWE compared with those without epilepsy, including anxiety (odds ratio [OR], 2.11; 95% CI 1.73–2.58), depression (OR, 2.45; 95% CI 1.94–3.09)

and alcohol dependence (OR, 4.94; 95% CI 3.50–6.96; Kwon et al., 2024). In sub-Saharan Africa, a systematic review of 16 health facility-based studies reported that the prevalence of comorbid depression in PWE ranged from 6.5 to 49.3% (Dessie et al., 2018). The systematic review, encompassing studies conducted in Ethiopia, demonstrated a pooled prevalence of depression (37.05%, 95% CI 28.23–45.86), anxiety disorders (33.81%, 95% CI 30.45–37.16) and CMDs (31.25%, 95% CI 17.00–45.49) in PWE (Tinsae et al., 2024).

Co-morbid CMDs in PWE have been associated with poorer seizure treatment outcomes and worse patient-reported health outcomes in high-income country settings (Gilliam et al., 2003; Josephson et al., 2017; Kanner, 2017; Mula et al., 2022). In these settings, there is robust, high-quality evidence that PWE and comorbid CMD have increased risk of poor seizure control (Josephson et al., 2017), premature mortality (Fazel et al., 2013; Gorton et al., 2021), anti-seizure medications side effects (Kanner, 2017), poor quality of life (QoL; Boylan et al., 2004; Gilliam et al., 2003; Jacoby and Baker, 2008; Jehi et al., 2011; Taylor et al., 2011) and increased functional disability (Centers for Disease Control Prevention, 2005; Sajobi et al., 2015). QoL is defined as "individuals' perceptions of their position in life, in particular to their goals, expectations, standards and concerns." It encompasses the individual social, emotional and cognitive function (Cramer et al., 1996). Functioning is described as "a difficulty in functioning at the body, person, or societal levels, in one or more life domains, as experienced by an individual with a health condition in interaction with contextual factors" (Garin et al., 2010b; Leonardi et al., 2006).

Comorbid CMDs and substance use have been associated with poor treatment adherence (Asghar et al., 2021; O'Rourke and O'Brien, 2017).

A systematic review and meta-analysis of 19 studies from low- and middle-income countries (LMICs) also found a significant negative association between comorbid depression (pooled effect size [ES] −1.16, 95% CI −1.70, −0.63) or anxiety (pooled ES −0.64, 95% CI −1.14, −0.13) on QoL of PWE (Tsigebrhan et al., 2023). However, all studies were cross-sectional and there was only one study reporting on the association between co-morbidity and functional disability. This evidence gap is important because the predictors of QoL or functional disability in PWE living in LMICs may differ due to the role of socio-economic factors and other variations in sociocultural context. The complex inter-relationships between emotional and social factors and QoL and functional disability have not been investigated in a rural, low-income African country. There have also been very few publications on the impact of comorbid substance use disorders on these important outcomes. Therefore, the research question for this study was "what is the impact of baseline co-morbid common mental disorder (CMD) symptoms and risky substance use on clinical outcome (seizure control), quality of life and functioning over a six month follow up period?"

The objective of this study was to investigate the impact of having comorbid CMD symptoms and/or risky substance use on QoL and functioning over a 6-month follow-up period. The study had the following hypothesis: CMD symptoms would, directly and indirectly, predict change in QoL and functional disability through the effect on seizure frequency.

## Methods

### Design

A primary healthcare-based prospective cohort study of PWE.

### Setting

The study was conducted in the Gurage zone in the Southern Nations, Nationalities and Peoples' Region of south-central Ethiopia. The Gurage zone is predominantly rural, characterised by fertile semi-mountainous terrain. Welkite town is its administrative centre. The study was conducted in four districts (Sodo, Eja, Wolikete and Kebena), with a total estimated population of 450,000–500,000 people. The Ethiopian healthcare system is divided into three tiers of service delivery. The first level consists of primary healthcare (PHC) units (health posts and PHC centres) and primary hospitals. PHC centres are generally staffed by nurses and health officers, serving a population of 25,000–40,000 people. Health posts are staffed by one or two community health extension workers (HEWs), serving a population of 3000–5000 people. Secondary-level services are provided by general hospitals and serve as referral centres from the primary hospitals; and tertiary-level services include specialised hospitals.

This study was nested in the scale-up phase of the *PR*ogramme for *Im*proving *Me*ntal Health Car*E* (PRIME) project which was a UK Department for International Development-funded research programme consortium across five LMIC (Ethiopia, South Africa, Uganda, India and Nepal; Lund et al., 2012). PRIME aimed to provide comprehensive evidence on how to integrate and scale up care for people with psychosis, depression, epilepsy, and/or alcohol use disorders. The focus was on integration in PHC settings using the World Health Organization's (WHO) mental health Gap Action Programme (mhGAP) intervention guide (Fekadu et al., 2016; Hailemariam et al., 2015; WHO, 2008). The programme of care was first implemented in the Sodo district (8 PHC centres) and, from 2016 onwards, scaled up to the other 14 districts in the Gurage zone (one PHC centre per district). The four study districts for the current study were selected purposively because of their high commitment to integrating mental healthcare and logistical considerations.

### Source population

The source population for this study was all people with a provisional diagnosis of convulsive epilepsy living in the four study districts of the Gurage zone.

### Screening and recruitment of study participants

Case detection was carried out by community key informants and HEWs who had been trained for two days to recognise people who may have active convulsive epilepsy, augmented by house-to-house screening by HEWs (Fekadu et al., 2016). Screen-positive individuals were referred to the nearby PHC centre and the diagnosis of epilepsy was confirmed by PHC workers who had been trained through PRIME and applied diagnostic algorithms outlined in the mhGAP intervention guide. This two-stage screening method has been used previously (Shibre et al., 2002) and was implemented in the PRIME study (Fekadu et al., 2016). The project psychiatric nurse then screened for eligibility, assessed for capacity to consent to participate in the study and obtained informed consent before a person was recruited into this cohort study.

*Inclusion criteria*: (a) PHC worker diagnosis of active convulsive epilepsy: two or more unprovoked convulsions separated by greater than 24 hours, with one convulsion taking place within the preceding 12 months (Mbuba et al., 2012; WHO, 2016); (b) aged

18 years or above and (c) no plans to out-migrate in the next 12 months.

*Exclusion criteria*: (a) Communication difficulties due to cognitive or intellectual disability; (b) unable to converse in Amharic, the official language of Ethiopia; and (c) lacking the capacity to consent after a psychiatric nurse assessment using the standardised approach used previously in this setting (Hanlon et al., 2016).

### Sample size determination

Based on a large, prospective study, the mean QoL score for PWE and depression was estimated to be 31.7 (standard deviation [SD] = 13.06) compared to 19.3 (SD = 13.87) in those without depression (Jehi et al., 2011). A total sample of 50 participants (25 with and 25 without co-morbid mental disorder) would be sufficient to detect this difference, with alpha 0.05 and power 0.8. To allow for the detection of a smaller difference in means (mean difference of 5.0), the required sample size was 88 in each group. To take account of clustering by district (n = 4), we assumed an intra-cluster correlation of 0.01 (Adams et al., 2004), resulting in a design effect of 1.21. Allowing for a 20% loss to follow-up, a total sample of 256 was required (128 per group).

### Measurement

Eligible people who gave informed consent to participate were interviewed at baseline ($T_0$), and again after 6 months ($T_1$) of follow-up. The hypothesised conceptual model is shown in Figure 1.

### Primary outcome ($T_0$ and $T_1$)

*QoL* was measured using the 10-item QoL in Epilepsy questionnaire (QOLIE-10-p; Cramer et al., 1996). This questionnaire was derived from the original 89-item version QOLIE-89 with an additional eleventh item to give a weighted total score (Cramer et al., 2000). The 10-item questionnaire has seven components: one item for each of five domains (seizure worry, overall QoL, emotional well-being, energy and cognitive functioning), two items on medication effects (physical effects and mental effects) and three items on social function (work, driving and social function). The total mean score ranges from 0 to 100 with a higher score indicating better QoL. For this study, the instrument was adapted and construct validity was established (Tsigebrhan et al., 2021a). The English version of the QOLIE-10-p was initially designed to be self-administered. For this study, the instrument was translated into Amharic, which is the local language, by the principal investigator and it was then back translated to English language by a non-mental health professional. The final Amharic version of the instrument was prepared after discussion of a group of psychiatrists with expertise in the area.

### Secondary outcomes ($T_0$ and $T_1$)

*Functional disability* was measured using the WHO Disability Assessment Schedule version 2.0 (12 item WHODAS-2; Üstün et al., 2010). The WHODAS-2 is a generic instrument that measures health-related functional disability in six domains of life during the previous 30 days. Each item is scored on a Likert scale starting from "no difficulty" 1 to "mild" 2, "moderate" 3, "severe" 4 or "extreme" difficulty 5. The recommended polytomous scoring method was used for analysis. A higher total score indicates a higher functional disability. WHODAS-2 has been validated in people with chronic diseases, including epilepsy (Garin et al., 2010a) and in Ethiopia (Habtamu et al., 2017). The 12 item even has shown superiority in understand ability and contextual relevance in a study done in people with severe mental disorders in the Ethiopian setting (Habtamu et al., 2017).

### Primary exposure ($T_0$ only)

*CMD (depression, anxiety, and somatic) symptoms*: The Self-Reported Questionnaire (SRQ-20) was developed by WHO to screen for CMD symptoms in the past 30 days (Beusenberg and Orley, 1994). The SRQ-20 items ask about depressive, anxiety, somatic symptoms and suicidal ideation. The total score is calculated by summing up all positive symptoms, ranging from 0 to 20. The SRQ-20 was previously translated into Amharic and validated in perinatal women and at PHC level (Hanlon et al. 2015, 2008a, 2008c; Kortmann and Ten Horn 1988). A score of eight was the optimum cut-off point for detection of depression at PHC level (Hanlon et al., 2015). For PWE in the same setting the optimum cut-off score of SRQ-20 indicating CMD was greater or equal to 9 (Tsigebrhan et al., 2021b).

*Substance use*: Risky use of alcohol, khat and tobacco was measured using the Alcohol, Smoking, and Substance Involvement Screening Test (ASSIST; WHO, 2002). The ASSIST has eight questions, with questions one to seven asking about use and problems related to substance use, and the eighth question inquiring about the use of injectable drugs. The total score for specific substance involvement is calculated by summation of the assigned numerical numbers from questions number 2–7 for each substance class. Low risk is indicated by a score of 0–10 for alcohol and 0–3 for

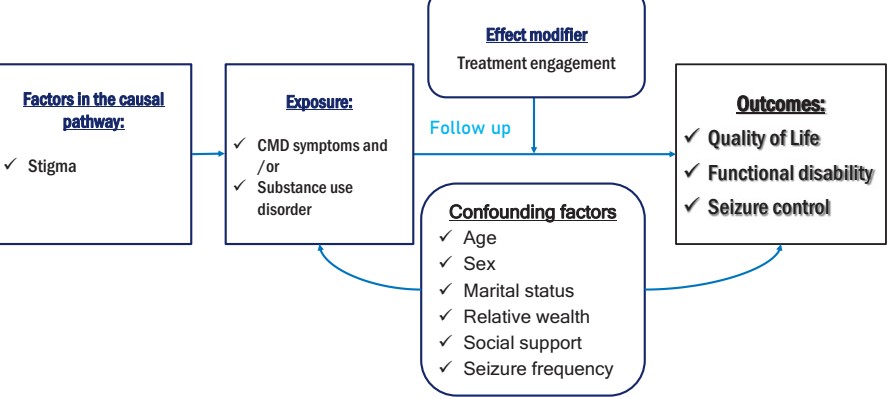

**Figure 1.** Conceptual model.

other substances, moderate risk is 11–26 for alcohol and 4–26 for other substances, and high risk is indicated by a score of 27 and above. The ASSIST has been contextually adapted in multiple countries including in Africa and Ethiopia (Ambaw et al., 2015; Humeniuk et al., 2008). For this study, the ASSIST was modified to assess commonly used substances in the southern part of Ethiopia: alcohol and khat (Fekadu et al., 2007; Schoenmaker et al., 2006).

### *Potential confounding variables ($T_0$ only)*

*Socio-demographic characteristics*: age, sex, education, marital status and income.

*Epilepsy-related factors*: duration of epilepsy and seizure frequency. At baseline, seizure frequency in the past 1 month was measured. At the follow-up time-point, the numbers of seizures in the last 6 months were recorded as follows: number per week (if <1/day), number per month (if <1/week) and number in the last 6 months (if <1/month). The severity of seizures was grouped into three categories based on their frequency in the past 6 months: seizure: –None, low to moderate (1–2 seizures) and high seizure severity (greater and equal to 3 seizures). This categorisation of seizure severity has been used in an African setting in a previous study (Fawale et al., 2014).

*Social support*: This was measured using the Oslo-3 item Social Support scale (OSSS-3; Dalgard et al., 2006). The OSSS-3 is a brief measurement of social functioning and has three items: The total score ranges from 3 to 14 with a higher score indicating better social support. OSSS-3 has been validated in an African setting (Abiola et al., 2013) and has been used in several studies in Ethiopia (Fekadu et al., 2014).

### *Factor hypothesised to be on the causal pathway*

Perceived stigma was measured using the stigma section of the Family Interview Schedule questionnaire (Sartorius and Janca, 1996). This instrument has been translated into Amharic and has been used previously in rural Ethiopia to measure stigma in PWE and their caregivers and those with mental disorders (Hanlon et al., 2008b; Shibre et al., 2006). Each item is rated in a four-point scale 0 "not at all," 1 "sometimes," 2 "often" and 3 "a lot" regarding the perceived stigma. A total score of one and above is considered as having the experience of perceived stigma.

### *Hypothesised effect modifier: epilepsy treatment engagement*

Treatment engagement was operationalised as the number of times the person attended the PHC centre in the preceding 6 months. Self-reported attendance was recorded and augmented by a medical record review. Good treatment engagement was defined as attending ≥4 times during the follow-up period.

### *Data collection and management*

All measures were carried out by experienced lay data collectors who have completed secondary school education. The lay data collectors were trained on the administration of the questionnaire for five days and practiced through role play before administering them to study participants. Immediately after the completion of data collection, the field supervisor checked the questionnaires for completeness. Data were double-entered using Epi-data version 3.1 (Lauritsen, 2004).

### Data analysis

Data were analysed using STATA version 17 (Hamilton, 2012). For continuous variables, indicators of central tendency were calculated depending on the distribution (mean with SD or median with interquartile range [IQR]). Percentages and frequencies were used to summarise categorical variables. Simple descriptive analyses were used to summarise the socio-demographic and clinical characteristics at $T_0$ and $T_1$. Wilcoxon ranked sum test or Fisher's exact test was used to examine the statistical significance of differences in baseline characteristics of those who were lost to follow-up and those who remained in the cohort. The dependent variables of change in QoL and change in functional disability were calculated by subtracting the total scores at $T_1$ from $T_0$.

Univariate and multivariable linear regression models were fitted to evaluate whether the primary exposure (comorbid CMD symptoms) predicted a change in the outcome variables (QOL and functional disability) adjusting for baseline outcome data. The pre-defined potential confounding variables (measured at the baseline) were also entered into the multivariable model. The risk of alcohol use was entered into the model separately from the total SRQ-20 score (CMD symptoms). Effect modification by number of PHC centre visits (treatment engagement) was tested using interaction terms with a total SRQ-20 score. A likelihood ratio test was used to examine statistical significance.

Structural equation modelling (SEM) was then conducted using R version 4.3 (Campbell and Campbell, 2019) to examine direct and indirect pathways through seizure frequency linking co-morbid CMD symptoms with QoL or functional disability. The direct and indirect pathways linking to the outcome were drawn based on the pre-hypothesised conceptual model (Supplementary file 1). Separate SEM was fitted for QoL and functional disability as two separate outcomes.

Before fitting the full SEM, CFA was carried out for each of the latent constructs of CMD symptoms, stigma, QoL and functional disability to examine the fit of the measurement models. The goodness of fit of the models was checked for each latent construct using the root mean square error approximation (RMSEA), Tucker–Lewis index (TLI) and comparative fit index (CFI). The significance of factor loadings of each item and the plausibility of the loadings were also examined. Weighted least square estimation was used for the complete data. The SEM was fitted again after multiple imputations of missing data using a chained equation (White et al., 2011).

### Results

### *Socio-demographic and clinical characteristics*

The study was conducted from March 2017 to June 2018. A total of 246 people were screened for active convulsive epilepsy but seven people did not fulfil the eligibility criteria and the data of two participants were incomplete.

At $T_0$, 237 participants were recruited. Of these, 92.4% (n = 219) were assessed after 6 months. There were two deaths and 16 participants could not be traced. Participants who were lost to follow-up were more likely to be single or previously married, had worse QoL, higher functional disability, had increased seizure frequency and more stressful life events compared to those who remained in the cohort (see Supplementary file 2).

Those participants who remained in the cohort had a median age of 32 years (IQR 22, 42), two-thirds were males (60.3%) and

**Table 1.** Characteristics of participants at $T_0$ (n = 237) and $T_1$ (n = 219; 6 months)

| Characteristics | | Baseline ($T_0$) n (%) | End line ($T_1$) n (%) |
|---|---|---|---|
| Age | In years | Median 30 (IQR 22, 42) | Median 32 (IQR 22, 42) |
| Sex | Male | 140 (59.1) | 132 (60.3) |
| | Female | 97 (40.1) | 87 (39.7) |
| Residence | Rural | 208 (87.8) | 193 (88.1) |
| | Urban | 29 (12.2) | 26 (11.9) |
| Education | No formal education | 135 (57.0) | 124 (56.6) |
| | Formal education | 102 (43.0) | 95 (43.4) |
| Marital status | Single, divorced or widowed | 114 (48.1) | 101 (53.9) |
| | Married | 123 (51.9) | 118 (46.1) |
| Relative wealth | Low or very low | 169 (71.3) | 155 (70.8) |
| | Average or above | 68 (28.7) | 64 (29.2) |
| Prescribed psychotropic medication (n = 207) | No | – | 190 (91.8) |
| | Yes | – | 17 (8.2) |
| CMD symptoms | Total SRQ–20 score | Median = 7 (IQR 3, 12) | Median = 3 (IQR 1, 7) |
| Risk of alcohol use (ASSIST score) | Low (ASSIST < 10) | 126 (67.4) | 147 (82.6) |
| | Moderate (ASSIST 11–26) | 34 (18.2) | 28 (15.7) |
| | High (ASSIST > 27) | 27 (14.4) | 3 (1.7) |
| Quality of life | Weighted QOLIE–10 score | Median 42.2 (IQR 28.7, 66.6) | Median 71.6 (IQR 45.8, 93.5) |
| Seizure frequency in the past 6 months | 0 | | 99 (45.2) |
| | 1 | | 87 (39.7) |
| | ≥2 | | 33 (15.1) |
| Social support | OSSS–3 total score | Mean 11.0 (SD 1.8) | Mean 11.2 (SD 1.39) |

ASSIST: Alcohol, Smoking and Substance Involvement Screening Test; OSSS: Oslo Social Support scale; QOLIE: Quality of Life in Epilepsy Questionnaire; SRQ-20: Self-Reported Questionnaire; WHODAS: World Health Organization Disability Assessment Schedule; CMD: common mental disorder.

56.6% had no formal education. Most of them (88.1%) resided in a rural area and nearly half (46.1%) were married (Table 1). All participants were diagnosed with generalised seizures, with a median of one seizure per month.

### Changes over the follow-up period

Over the 6 month follow-up period, participants attended the PHC centre a median of 5 times (IQR 5–6) for epilepsy and/or mental healthcare. The median score for CMD symptoms and the risk of alcohol use decreased from baseline to the 6 months follow-up assessment (Table 1). There was a positive change in QoL (mean

QOLIE-10p score = 18.92 (SD = 38.19)) and improvement in functional disability (mean = −6.77; SD = −19.11).

### Status at 6 months

At 6 months, almost half of the participants (45.2%) were seizure-free. Almost all (n = 189, 90%) were taking one anti-seizure medication (phenobarbitone) and 10% (n = 21) were taking two (phenobarbitone plus either carbamazepine or valproate). Only 8.2% (n = 17) were on any psychotropic medication.

### Regression analysis: QoL

CMD symptoms were not significantly associated with a change in the QoL in the crude or adjusted regression analysis (adjusted β coef. = −0.37, 95% CI −1.30, 0.55; Table 2). Seizure frequency was significantly associated with a decreased change in the QoL in the multivariable model (β coef. = −1.73, 95% CI −2.73, −0.74). When the risk of alcohol use was entered into the multivariable model instead of the SRQ-20 score, there was no significant association between moderate to high-risk alcohol use and change in the QoL (β coef. = −0.70, 95% CI −9.20, +7.81) compared to low-risk alcohol use.

Those participants who had good treatment engagement had a better change in QoL than those with poor treatment engagement (β coef. = 14.6, 95% CI 3.70, 25.51) in the univariable analysis. Treatment engagement did not significantly modify the association between CMD symptoms (SRQ-20 score) and QoL (interaction coefficient = 1.03, 95% CI −0.93, 3.0; Likelihood ratio test $\chi^2$ = 3.48, p = 0.18).

### Regression analysis: functional disability

CMD symptoms were not significantly associated with a change in functional disability (β coef. = 0.03, 95% CI −0.48, +0.54). Increased seizure frequency was the only factor significantly associated with a change in functional disability in both univariable and multivariable analysis (adjusted β coef. = +0.88, +0.32, +1.44; see Table 2). When the risk of alcohol use was entered into the multivariable model instead of SRQ-20 total score, there was no significant association between moderate to high-risk alcohol use and change in functional disability (β coef. = −1.31, 95% CI −5.89, 3.26) compared to low-risk alcohol use.

Those participants who had good healthcare engagement (≥4 health centre attendance) had a better change in their disability score than those with poor attendance (β coef. = −8.13, 95% CI −14.01, −2.24) in the univariable analysis. Healthcare engagement was not an effect modifier of the association between CMD symptoms (SRQ-20 score) and functional disability (interaction coef. = −0.44, 95% CI −1.50, +0.62; Likelihood ratio test: $\chi^2$ = 4.65, p = 0.10).

### SEM: QoL

The fit indices for each measurement model (stigma, CMD symptoms, social support, and QoL) indicated adequate fit to the data (Supplementary file 3). The fit indices for the full structural model also indicated adequate fit of the model to the data ($\chi^2$ = 1554.2; degree of freedom = 1072), p < 0.0001, CFI = 0.97, TLI = 0.97 and RMSEA = 0.06. In the full SEM, QoL at $T_1$ was significantly predicted by seizure frequency in the 6 month follow-up period

**Table 2.** Univariable and multivariable regression analysis of factors associated with a change in quality of life score/change in functional disability between $T_1$ and $T_0$ (6 months)

| Characteristics | | Change in quality of life | | Change in functional disability | |
|---|---|---|---|---|---|
| | | Crude β coef. (95% CI) | Adj. β coef. (95% CI) | Crude β coef. (95% CI) | Adj. β coef. (95% CI) |
| CMD symptoms (total SRQ–20 score at baseline) | | −0.79 (−1.67, +0.09) | −0.37 (−1.30, +0.55) | 0.23 (−0.26, +0.72) | 0.03 (−0.48, +0.54) |
| Sex | Female | −5.60 (−12.99, +1.79) | −4.36 (−12.0, +3.28) | +3.10 (−0.89, +7.11) | +3.31 (−0.80, +7.41) |
| Age (years) | | −0.02 (−0.31, +0.26) | −0.04(−0.41, +0.33) | +0.13 (−0.10, +0.29) | +0.13 (−0.07, 0.33) |
| Education | No formal | 1 | 1 | 1 | |
| | Formal | −0.45 (−7.79, +6.89) | −2.96 (−10.64,+4.72) | −1.50 (−5.46, +2.46) | +1.36 (−2.81, +5.54) |
| Relative wealth | Average or above | 1 | 1 | 1 | |
| | Low or very low | +3.75 (−4.39, +11.88) | +2.71 (−5.37, +10.80) | −0.14 (−4.46, +4.19) | −0.50 (−4.88, +3.88) |
| Marital status | Married | 1 | 1 | 1 | |
| | Single or formerly married | +3.45 (−3.85, 10.75) | +7.25 (−1.23, 15.73) | −4.36 (−8.26, −0.46) | −3.97 (−8.63, +0.69) |
| Duration of epilepsy (years) | | −0.14 (−0.51, +0.23) | −0.23(−0.62, +0.15) | +0.15 (−0.04, +0.35) | +0.13 (−0.08, +0.34) |
| Seizure frequency (month) | | −1.78 (−2.63, −0.93) | −1.73 (−2.73, −0.74) | 0.84 (0.28, 1.39) | 0.88 (0.32, 1.44) |
| OSSS score | | +1.08(−0.98, +3.14) | +1.29 (−0.74, +3.33) | −0.54 (−1.65, +0.57) | −0.59 (−1.69, +0.50) |

CMD: common mental disorder; OSSS: Oslo Social Support scale; QOLIE: Quality of Life in Epilepsy Questionnaire; SRQ-20: Self-Reported Questionnaire; WHODAS: World Health Organization Disability Assessment Schedule.

(B = −0.91, 95% CI −1.16, −0.66) but not by $T_0$ CMD symptoms directly (B = −0.14, 95% CI −0.31, +0.030) or indirectly through the seizure frequency (B = −0.12, 95% CI −0.26, +0.013). CMD did not have a significant effect on seizure frequency (B = 0.14, 95% CI −0.015, +0.29). However, the summative (direct + indirect) effect of CMD on QoL was significant (B = −0.27, 95% CI −0.48, −0.056). Baseline stigma (B = 0.83, 95% CI 0.64, 1.03) was a significant predictor of CMD symptoms (Figure 2).

### SEM: functional disability

The fit indices for the full structural model indicated adequate fit of the data by $\chi^2$ = 1580 (degree of freedom = 1167), $p < 0.0001$, CFI = 0.95, TLI = 0.99 and RMSEA = 0.06. Functional disability at $T_1$ was predicted by baseline ($T_0$) CMD symptoms (B = 0.24, 95%

CI 0.06, 0.41) and seizure frequency (B = 0.67, 95% CI 0.46, 0.87; Figure 3). Seizure frequency (B = 0.09, 95% CI −0.01, +0.05) did not have a mediation effect on the relationship between CMD symptoms and functional disability. The summative (direct plus indirect) effect of CMD symptoms on functional disability was significant (B = 0.34, 95% CI 0.14, 0.52).

### *Sensitivity analysis*

Similar model fit indices were obtained after imputation of missing data. However, in the imputed model, CMD symptoms directly predicted seizure frequency (B = 0.17, 95% CI 0.3, 0.31), and the indirect (B = −0.15, 95% CI −0.27, −0.03) and total effect B = −0.28, 95% CI −0.48, −0.07) of CMD symptoms on QoL through seizure frequency also became significant (see Supplementary file 4).

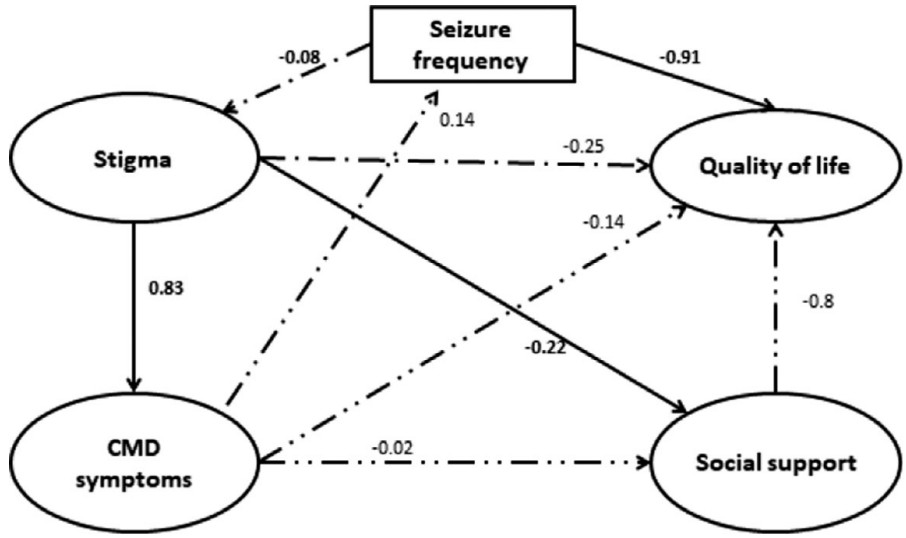

**Figure 2.** Structural equation model of end line quality of life regressed onto the latent constructs of baseline stigma, CMD symptoms and social support.

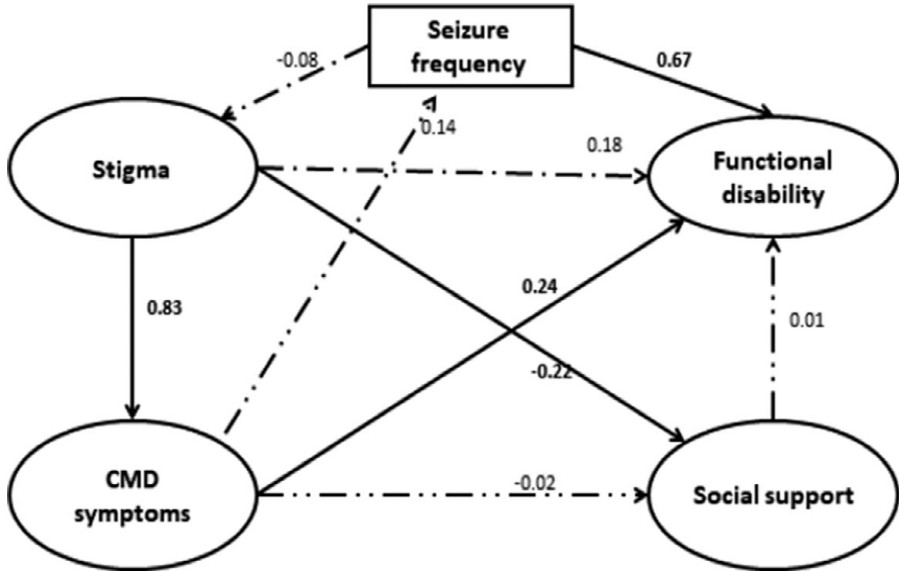

**Figure 3.** Structural equation model of end-line functional disability regressed on the latent construct of baseline stigma, CMD symptoms and social support.

## Discussion

In this prospective cohort study, we investigated the impact of having comorbid CMD symptoms in PWE living in rural Ethiopia on QoL and functional disability. In hypothesis-driven regression analyses, neither baseline CMD nor risky alcohol use were associated with a change in functional disability or QoL, nor moderated by treatment engagement. However, SEM indicated that baseline CMD had a significant direct impact on functional disability at follow-up. Only the summative effect of CMD on QoL was significant.

The lack of a prospective association between co-morbid CMD symptoms and change in QoL (in the linear regression model) contrasted with the SEM finding of a significant summative effect of baseline CMD on QoL at 6 months. The SEM complete case analysis did not find CMD to be associated either directly or indirectly (via seizure control) but sensitivity analysis with multiple imputations of missing data indicated that CMD affected QoL through the mediator of seizure frequency. Our study was likely to have been underpowered and affected by attrition bias which may mean the findings from the multiple imputation analysis are more valid. Cross-sectional analyses of the same cohort at baseline (Tsigebrhan et al., 2021a) and cross-sectional studies of the association in other LMIC settings (Tsigebrhan et al., 2023) showed strong associations between CMD and QoL but are more susceptible to negative recall bias (Katschnig, 2006) than prospective studies and do not illuminate the potential mechanism of any association and, indeed, its temporal relationship. Furthermore, CMD symptoms may have been managed by PHC workers between baseline and follow-up, supported by the reduced total score of SRQ-20 over time, although there was no evidence of effect modification by treatment engagement.

The association of increased seizure frequency with poor QoL is consistent with the results of studies from high-income countries and from Africa (Addis et al., 2021; Ogundare et al., 2020; Taylor et al., 2011). As QoL measurement was also related to the subjective experience of being satisfied and fulfilled in life (Katschnig, 2006), the direct social and cultural effect of increased seizure frequency on their overall life could be the most troublesome problem for these participants. The SEM sensitivity analysis indicated that seizure frequency may mediate the association between CMD symptoms and QoL and the direct association between CMD and seizure frequency was significant. Previous studies have shown that people with CMD symptoms are less likely to be seizure free (Josephson et al., 2017; Medel-Matus et al., 2022). Common mental health conditions like depression have been found to contribute to treatment resistance epilepsy (Medel-Matus et al., 2022), poor treatment adherence (Asghar et al., 2021; O'Rourke and O'Brien, 2017) and increased anti-seizure medication side effects (Kanner, 2017). Therefore, comorbid CMD symptoms could have directly contributed to poor anti-seizure medication adherence and side effects which then affected achieving seizure control. Unfortunately, these factors (adherence and anti-seizure medication side effects) were not measured in our study which has limited our findings. We found that only half of the participants were seizure-free at the end of the cohort rather than 70% which is expected for the first-line treatment of generalised tonic-clonic seizure with anti-seizure medication (Kanner and Bicchi, 2022). Beyond the potential impacts of CMD symptoms, this may also reflect the scarcity and high cost of the alternative classes of anti-seizure medications in this low socio-economic status setting.

For the outcome of functioning, there was also a discrepancy between findings from the linear regression and SEM. However, SEM provided strong evidence of a direct effect of co-morbid CMD symptoms on functional impairment. The global burden and disability associated with depression is substantial (Lépine and Briley, 2011), compounding disability associated with the underlying chronic neurologic disorder (epilepsy). Meeting basic needs, like food and shelter, is often given highest value by people with chronic mental health conditions in the same setting (Mall et al., 2017). Therefore, being functional and thus better able to meet basic needs could be more important than satisfaction with life and could explain the stronger prospective associations between CMD and functional disability compared to QoL. The impact of seizure frequency on functional disability was also significant, in keeping with other studies (Centers for Disease Control Prevention, 2005; Sajobi et al., 2015), but there was no evidence of CMD symptoms indirectly affecting functioning through seizure frequency similar to QoL.

There was also significant association of epilepsy-related stigma and CMD symptoms on the SEM analysis.

Risky alcohol use was not associated with a change in QoL or functioning. Levels of risky alcohol use were high at baseline, with 14.4% of PWE having high-risk use of alcohol. This decreased substantially (to 1.7%) over the 6-month follow-up period and could explain why baseline risky alcohol use was not associated with either outcome. At baseline alcohol withdrawal could have been the primary cause of seizures (and/or epilepsy) or alcohol use disorder could be comorbid with epilepsy (Gorton et al., 2021). Evidence from high-income countries indicates a higher prevalence of alcohol use in PWE compared to the general population (Lu et al., 2021) which is associated with a higher rate of mortality of PWE (Fazel et al., 2013; Gorton et al., 2021).

## Strength and limitations

To the best of our knowledge, this study was the first in Ethiopia or any other low-income country setting to investigate prospectively the impact of comorbid mental health conditions in PWE on QoL and functional disability. The setting reflected the normal routine care of PWE at PHC level in contrast to the many studies based in tertiary referral facilities. Alongside these strengths, there were however some limitations of the study. Even though the epilepsy definitions used by WHO's mhGAP and International League Against Epilepsy are similar, diagnostic tools like EEG were not used in this study. Though the percentage of people who were lost to follow-up was minimal (7%), there was evidence of selective attrition by people who had higher CMD symptoms at baseline and differences in the final result of the SEM between the complete and imputed data. We were not able to recruit to the proposed sample size and the analysis was underpowered. Treatment engagement is a complex and multi-dimensional construct (Lindsey et al., 2014), and the attitudinal and behavioural component was not measured in this study. This, alongside the sample size, could explain the absence of significant effect modification by treatment engagement in the association between CMD symptoms and the outcomes. We were not able to assess the cognitive function of the participants which could have contributed to poor QoL and decreased functioning.

In conclusion, comorbid CMD symptoms and seizure frequency had independent negative impacts on functional disability. Seizure frequency also predicted poor QoL and the sensitivity analyses indicated a possible mechanism linking CMD symptoms with poor QoL through seizure frequency. Therefore, the enhancement of integrated mental and physical care for individuals with epilepsy necessitates the inclusion of routine mental health screening performed by healthcare professionals and the effective treatment of highly prevalent comorbid mental health conditions, such as depression. Though pharmacological management was frequently practiced by the PHC workers, the social and emotional recovery of PWE in this context tends to be neglected (Catalao et al., 2018). Hence, it is highly recommended for clinicians to examine the number of psychosocial problems contributing to poor mental health of PWE alongside the prescription of anti-seizure medications. Cost-effective psychosocial interventions delivered by non-mental health specialists could also be beneficial in the management of common mental health conditions (Singla et al., 2017). Future research with a larger sample size and longer periods of follow-up are needed to examine the association of comorbid mental health conditions and QoL. Research on interventions that address mental, social and physical health adversities should be adapted,

implemented and evaluated in this rural community. The availability and sustainable provision of not only the older anti-seizure medications but also the newly available anti-seizure medications is important to achieve good control of seizures and thus improve QoL and functioning. Stigma reduction programs and interventions at the community level are also highly recommended to support social inclusion of PWE and minimize the impact on mental health (Chakraborty et al., 2021).

**Open peer review.** To view the open peer review materials for this article, please visit http://doi.org/10.1017/gmh.2025.24.

**Supplementary material.** The supplementary material for this article can be found at http://doi.org/10.1017/gmh.2025.24.

**Data availability statement.** The datasets used and/or analysed during the current study are included in the article as Supplementary file 5.

**Acknowledgements.** We are grateful for the participants and their families, the PRIME project and its entire staff.

**Author contribution.** RT, CH and CN participated in the writing of the research proposal. RT contributed to the collection of the data. RT, GM, MB and CH analysed the data. RT drafted the manuscript. RT, CH, GM, MB and CN made an intellectual contribution and revised the draft. All the authors have read and approved the final manuscript.

**Financial support.** This study was conducted as part of a Wellcome Trust fellowship for RT (Grant Number 104023/Z/14/A) and a PhD fellowship from Centre for Innovative Drug Development and Therapeutic Trials for Africa (CDT-Africa). CH receives support from the National Institute for Health and Care Research (NIHR) through the NIHR Global Health Research Group on Homelessness and Mental Health in Africa (NIHR134325) using UK aid from the UK Government. The views expressed in this publication are those of the authors and not necessarily those of the NIHR or the Department of Health and Social Care. CH also receives support from the Wellcome Trust through grants 222154/Z20/Z and 223615/Z/21/Z. For the purposes of open access, the author has applied a Creative Commons Attribution (CC BY) licence to any Accepted Author Manuscript version arising from this submission.

**Competing interest.** The authors declare that they have no competing interests.

**Ethics statement.** All the methods were performed in accordance with the Declaration of Helsinki. Ethical approval was obtained from the Institutional Review Board of the College of Health Sciences, Addis Ababa University and the Research Ethics Committee of King's College London (HR-15/16-2434). Informed consent and witnessed verbal consent (for non-literate participants) were sought after adequate information was provided.

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
