## [Reviewer Report]

This is a commendable prospective cohort study to provide scientific evidence for practice, and in line with the global effort at providing holistic care, in medical practice especially, even in PHC setting.

The study objectives are clearly stated backed with sound methodology. The measurements and data analysis are appropriate to the objectives of the study.

The author’s demonstrated the utility of the structural equation modelling (SEM) to overcome possible limitations of traditional regression analyses in demonstrating direct association between variables (especially in lower powered studies), to reveal significant indirect effects of possible confounders.

The prospective design of study is also a strong point over findings of previous cross sectional studies, with possible interactive effect of therapeutic engagement during follow up period.

Authors acknowledged the diagnostic and therapeutic limitations among study participants which are in keeping with the socioeconomic and PHC setting of study.

---

## [Reviewer Report]

This study lie in its methodological rigor, focus on an under-researched population, identification of key predictors, and alignment with healthcare initiatives aimed at improving mental health services. These factors collectively enhance the study’s contribution to understanding the complex relationships between epilepsy, mental health, and functional outcomes in a rural Ethiopian context

Abstract: Refrain from using abbreviations like CMD and QOL.

Introduction

1. Clarify Objectives:

The introduction should explicitly state the study’s objectives and hypotheses more clearly. While the background provides context, a concise statement summarizing the main research questions would enhance clarity.

2. Strengthen Literature Review:

Expand the literature review to include more recent studies that highlight gaps in existing research, particularly focusing on CMDs and epilepsy in LMICs. This would provide a stronger foundation for the study’s relevance.

3. Define Key Terms:

Ensure that key terms such as “common mental disorders,” “quality of life,” and “functional disability” are clearly defined early in the introduction to avoid ambiguity later in the manuscript.

Methods

1. Study Design – The study does not include a non-exposed group, which means it is not a cohort study. Instead, it is a prospective follow-up study or a single cohort study.

2. Participant Recruitment:

Clarify the recruitment process for participants, including how many individuals were screened versus how many were included in the final analysis. This transparency is crucial for understanding selection bias and generalizability.

Discussion

1. Implications for Practice:

Clearly outline practical implications for healthcare providers based on findings, emphasizing how integrated care approaches can be developed or improved based on study results.

Limitations and Future Research:

A dedicated section addressing limitations should be included, discussing potential biases, confounding factors, and areas for future research to further investigate these relationships.

---

## [Editor Report]

The impact statement, abstract, methods including tools and the results are well and satisfactory stated. 

On line 202/203 need to separate the names.

The discussion is too long and could be made more concise. Most of the half of the discussion is re-statement of results which could be easily discussed and incorporated and, in the process, substantially reduce the length of the discussion in the last section, starting from line 448.